# Sustainable Protein Recovery and Wastewater Valorization in Shrimp Processing by Ultrafiltration

**DOI:** 10.3390/foods14122044

**Published:** 2025-06-10

**Authors:** Samar Ltaief, Aurélie Matéos, Adrien Forestier, Khaled Walha, Loubna Firdaous

**Affiliations:** 1UMR Transfrontalière BioEcoAgro N° 1158, Université de Lille, INRAE, Univ. Liège, UPJV, YNCREA, Univ. Artois, Univ. Littoral Côte d’Opale, ICV–Institut Charles Viollette, F-59000 Lille, France; samar.ltaief.etu@univ-lille.fr (S.L.); aurelie.mateos@univ-artois.fr (A.M.); adrien.forestier@univ-lille.fr (A.F.); 2Laboratoire Sciences des Matériaux et Environnement, Faculté des Sciences de Sfax, Université de Sfax, Sfax 3000, Tunisia; khaled.walha@fss.usf.tn

**Keywords:** shrimp processing wastewater, protein recovery, ultrafiltration, wastewater valorization, seafood sustainability

## Abstract

This study investigated the use of ultrafiltration for sustainable protein recovery and the treatment of shrimp washing wastewater (SWW). Three ultrafiltration membranes with molecular weight cut-offs of 5, 10, and 50 kDa were tested using a combined ultrafiltration–diafiltration process (UF-DF). The performance of each membrane was assessed based on protein recovery efficiency, chemical oxygen demand (COD) reduction, turbidity, fouling behavior, and cleaning efficiency. The 5 kDa membrane showed superior performance, achieving over 90% protein and COD rejection and producing the highest protein-enriched retentate. It also exhibited the lowest fouling index and best cleaning recovery, confirming its suitability for protein concentration and wastewater treatment. This research highlights UF-DF as a promising, eco-efficient technology for valorizing seafood processing effluents by recovering high-value proteins and reducing environmental discharge loads.

## 1. Introduction

Despite its economic significance, the shrimp processing industry produces a lot of waste, which represents a substantial environmental challenge and results in the loss of valuable resources. These wastes can be roughly divided into two categories: solid wastes, such as heads, shells, and exoskeletons, and liquid wastes, including washing waters and process effluents [1]. Rich in proteins, chitin, and carotenoids, solid by-products have attracted increasing attention for their value-added applications such as chitosan production, animal feed, and biofertilizers [2]. However, liquid wastes remain underutilized and environmentally problematic due to their high volumes and pollutant loads [3]. Washing operations alone can consume 10 to 40 m^3^ of water per ton of raw shrimp, generating effluents rich in organic matter, fats, and nutrients [4]. Compared to other agro-industrial effluents, shrimp wastewater is uniquely challenging because it contains high concentrations of proteins, lipids, and carbohydrates from shrimp tissue and processing aids, as well as high levels of nitrogen and phosphorus from protein degradation and phosphate additives [5,6,7]. Additionally, it may contain chitin fragments and exhibit high salinity, especially in marine shrimp processing, further complicating treatment and disposal. These factors contribute to an extremely high chemical oxygen demand (COD), which depletes oxygen in receiving waters, and high turbidity and nutrient loads that hinder photosynthesis and promote eutrophication [6,7,8].

Traditional biological and physicochemical treatment methods are generally used to reduce the environmental impact of seafood effluents. However, these methods often involve high costs due to necessary pretreatment and multi-stage processes. Moreover, they often fail to recover valuable biocomponents or meet strict discharge standards [9]. Recovering valuable organic matter from crustacean wastewater offers a way to reduce treatment costs while enabling waste utilization and the extraction of bioactive compounds. Indeed, seafood effluents are rich sources of high-value compounds such as proteins, essential amino acids, PUFAs, vitamins, and minerals [10,11,12,13], which have significant potential in various industries, including food, nutraceuticals, pharmaceuticals, and agriculture.

Ultrafiltration (UF) represents a promising approach for recovering these valuable molecules from seafood processing wastewater [14]. Through selective compound retention, UF is a sustainable and cost-effective technology that lowers pollution without the need for chemicals or absorbents. In addition to producing high-quality permeate that can be used again in the processing facility, this process enables the simultaneous concentration and purification of valuable components, such as proteins.

UF has been successfully used in various water treatment applications [15] and has demonstrated its effectiveness in recovering and concentrating valuable compounds from marine by-products [16]. For instance, Afonso et al. [17] used UF to treat wastewater from seafood processing, successfully recovering protein content and enabling the reuse of treated water. Previous research highlights the potential of ultrafiltration to treat shrimp processing wastewater and recover valuable biomolecules from byproducts. For example, Amado et al. [12] successfully recovered astaxanthin and bioactive peptides using a 300 kDa membrane. In another study [13], they showed that UF membranes (30–300 kDa) achieved high protein recovery rates, with the 30 kDa membrane recovering over 80% of proteins and concentrating them seven-fold. Ferjani et al. [18] reported significant COD removal (50–65%) using a cellulose acetate membrane for wastewater treatment in seafood processing. Similarly, Oliveira et al. [19] used a UF-based filtration system to reduce organic matter in surimi washing waters, successfully concentrating proteins, essential amino acids, and carotenoids for potential applications in the food industry.

To further enhance selectivity and sustainability, diafiltration (DF) can be integrated with UF. DF involves the addition of clean water to the retentate during filtration, which helps to remove smaller undesirable solutes like salts, small organics, and low-molecular-weight impurities. This results in a higher purity and concentration of target macromolecules like proteins. Importantly, by reducing the contaminant load in the final permeate and enhancing retentate quality, DF supports circular resource recovery and environmental protection [20,21]. For example, Li et al. [22] demonstrated this by effectively obtaining a protease-rich concentrate from tuna spleen extract using a UF-DF sequence with a 30 kDa membrane. Similarly, Tonon et al. [23] further illustrated the benefits of UF pretreatment for concentrating shrimp cooking water prior to enzymatic hydrolysis, resulting in a fraction rich in essential amino acids and antioxidants. However, despite growing evidence of its effectiveness in seafood processing, the combined UF-DF approach is still not explored for the treatment of shrimp washing wastewater specifically, a rich but under-exploited waste stream.

This study aims to fill this significant gap by evaluating the effectiveness of an ultrafiltration–diafiltration (UF-DF) process in recovering proteins from shrimp washing wastewater. By comparing the performance of three ultrafiltration membranes with different molecular weight cut-offs, we aim to identify the optimal MWCO for maximizing protein retention and minimizing environmental discharge loads. Special attention is given to membrane fouling mechanisms, flux behavior, and the quality of both the permeate and retentate streams. The rationale behind this research lies in the dual objective of environmental protection and resource recovery.

## 2. Materials and Methods

### 2.1. Wastewater Samples

The wastewater samples were kindly provided by a seafood processing company located in the city of Sfax, southern Tunisia. The wastewater was collected and transported under frozen conditions to the laboratory. Upon arrival, subsamples were taken for analysis, and the remaining portions were stored at −18 °C until further use.

### 2.2. Ultrafiltration–Diafiltration (UF-DF) Procedure

Ultrafiltration experiments were conducted at the laboratory scale using a LABCELL CF-1 system (Koch Membrane Systems, Stafford, UK) equipped with a flat-sheet membrane cell (75 mm diameter, effective filtration area: 28 cm^2^). The setup included a 500 mL double-walled feed tank, a recirculation pump, a pressure gauge (1–10 bar), and a pressure regulation valve. A continuous flow of tap water through the tank’s double wall maintained a constant feed temperature, which was monitored via an in-tank thermometer.

HyStream TangenX membranes (REPLIGEN, Waltham, MA, USA) with nominal molecular weight cut-offs (MWCOs) of 5, 10, and 50 kDa were employed. Prior to each filtration run, membranes were characterized for their pure water flux (*J_w0_*) under variable transmembrane pressures (ΔP), and their initial hydraulic permeability (*L_p0_*) was determined from a linear regression of *J_w0_* versus ΔP using the following equation:(1)$$J_{w0}=L_{p0}\times ∆P=\frac{∆P}{\mu R_{m}}$$
where *J_w0_* is the water permeation flux (m^3^·m^−2^·s^−1^), *L_p0_* is the pure water permeability (m.s^−1^·Pa^−1^), µ is the water viscosity (Pa·s), and *R_m_* is the intrinsic membrane resistance (m^−1^).

Ultrafiltration–diafiltration tests were conducted in the concentration mode at a constant transmembrane pressure of 2 bar, with a crossflow velocity of 1.7 m·s^−1^ and a temperature of 20 °C. Starting with 500 mL of feed, ultrafiltration proceeded until a volume reduction factor (VRF) of 5 was reached. This was followed by diafiltration, during which the retentate was diluted with four diavolumes (DVs) of deionized water and reconcentrated to the same VRF. One diavolume (DV) corresponds to the volume of deionized water added to the retentate during diafiltration that is equal to the initial volume of the retentate before dilution. This UF-DF sequence was repeated once, yielding two complete diafiltration cycles. Samples of permeates and retentates were collected at each step for pH and conductivity measurements and stored at 4 °C for further analyses.

After each UF-DF sequence, the system was emptied, and membrane cleaning was performed to restore flux performance. A preliminary rinsing phase using distilled water was conducted for 10 min at 2 bar and 20 °C, during which the retentate stream was discarded to avoid redeposition of foulants. Post-rinse water permeability (*L_p1_*) was then measured.

To fully recover the membrane’s initial performance, a chemical cleaning step followed: a 0.02 M NaOH solution containing 3 mL of 3.6% NaClO was recirculated (500 mL) at 40 °C for 90 min under 2 bar. Final rinsing with deionized water continued until a neutral pH was achieved. Final membrane permeability (L_p2_) was assessed, and the cleaning efficiency (CE) and fouling index (FI) were calculated as follows:
(2)$$CE=\left(\frac{L_{p2}}{L_{p0}}\right)\times 100$$
where *L_p2_* is the water permeability after cleaning (m·Pa^−^^1^·s^−^^1^) and *L_p0_* is the water permeability of the new membrane (m·Pa^−^^1^·s^−^^1^). The fouling index (*FI*) was also assessed by comparing the water permeation flux before and after the ultrafiltration:
(3)$$FI=\left(1-\frac{L_{p1}}{L_{p0}}\right)\times 100$$
where *L_p1_* is the water permeation flux after ultrafiltration (m·Pa^−1^·s^−1^).

All ultrafiltration experiments were performed in triplicate using new membranes for each replicate. Results are reported as mean ± standard deviation. While no analysis of variance (ANOVA) was conducted, trends across replicates were consistent.

### 2.3. Analytical Methods

#### 2.3.1. pH, Conductivity, and Turbidity Measurements

All measuring devices were calibrated according to the manufacturers’ instructions before each use. pH, conductivity, and turbidity were measured using a Mettler Toledo FE20 pH meter, a Mettler Toledo F30 conductivity meter, and a HANNA HI98317 turbidity meter, respectively.

#### 2.3.2. Dry Matter and Ash Content

The dry matter content was determined by oven-drying the samples at 105 °C until a constant weight was achieved. The ash content was determined by first drying the samples in a ceramic crucible at 105 °C for three hours and then heating them overnight in a muffle furnace at 550 °C. The crucible was then cooled in a desiccator at room temperature for 30 min before weighing.

#### 2.3.3. Chemical Oxygen Demand

COD was measured in feed and permeate samples to evaluate the organic load in the shrimp washing wastewater before and after treatment. This parameter is crucial for assessing how well the ultrafiltration–diafiltration process reduces environmental pollution. Samples underwent chemical digestion at 150 °C for 90 min using a Stuart SBH200D/3 block heater. COD was determined by measuring absorbance at 565 nm with a WTW™ photoLab™ spectrophotometer (WTW™, Paris, France).

#### 2.3.4. Protein Concentration

Protein concentration was determined using a Bicinchoninic Acid (BCA) Protein Assay Kit, using bovine serum albumin (BSA) as a standard. Briefly, 25 µL of each sample was mixed with 200 µL of working reagent in a 96-well microplate. The plate was covered and incubated at 37 °C for 30 min. After cooling to room temperature, absorbance was measured at 562 nm using a SpectraMax^®^ ABS Plus microplate reader. A standard curve was prepared using BSA concentrations ranging from 0.062 to 2 g·L^−1^ (R^2^ = 0.9964). Samples with absorbance values above 2 were diluted and re-analyzed. All analyses were performed in triplicate on feed, concentrate, and permeate samples. Average values are given.

### 2.4. SDS-PAGE Electrophoresis

The protein fractions contained in initial washing waters, retentates, and permeates obtained by UF-DF were characterized by polyacrylamide gel electrophoresis in the presence of sodium dodecyl sulfate (SDS-PAGE). Electrophoresis was carried out following the method defined by Laemmli (1970) [24] with slight modifications.

The SDS-PAGE resolving gel was a 15% (*w*/*v*) polyacrylamide gel containing 375 mM of Tris-HCl buffer at pH 8.8 and 0.1% (*w*/*v*) SDS. The stacking gel was composed of 4% (*w*/*v*) polyacrylamide with 125mM Tris-HCl buffer at pH 6.8 and 0.1% (*w*/*v*) SDS.

According to the BCA protein assay kit’s protein concentration results, all samples had a low protein content (<2 µg·µL^−1^). As a result, all samples were half-diluted using 125 mM Tris-HCl buffer, pH 6.8, containing 0.2% (*w*/*v*) SDS, 5% (*v*/*v*) β-mercaptoethanol, 10% (*v*/*v*) glycerol, and 0.01% (*w*/*v*) bromophenol blue (100 µL of sample with 100 µL of buffer). Each sample was loaded into the gel with a volume of 15 µL after being heated for 10 min at 95 °C. Separation was performed at a constant voltage of 120 V for 60 min using the Mini-Protean II system (BioRad Laboratories, Hercules, CA, USA). The gels were subjected to overnight staining with InstantBlue™ (Expedeon, Cambridgeshire, UK) and subsequently washed with distilled water until the background color vanished. As a molecular weight standard, a Precision Plus Protein Dual Color Standard (BioRad Laboratories, Hercules, CA, USA) (10–250 kDa) was used. The SDS-PAGE gels were then scanned with a Bio-Rad Gelloc-calibrated densitometer (Hercules, CA, USA).

## 3. Results and Discussion

### 3.1. Shrimp Washing Wastewater (SWW) Characterization

Shrimp washing wastewater samples before and after vacuum filtration were first characterized in terms of conductivity, ash, turbidity, COD, and protein content (Table 1). All measurements were performed in triplicate, and values are expressed as mean ± standard deviation. In general, the SWW exhibited a cloudy yellowish–orange appearance, attributed to the presence of astaxanthin, with visible suspended particles, resulting in a high turbidity of approximately 593 NTU. As shown in Table 1, the SWW was characterized by high organic content, as evidenced by its COD and protein content, along with significant dissolved and suspended solids loading, as indicated by its conductivity, turbidity, and dry matter content. The pH value of the SWW (8.2 ± 0.2) revealed a slightly alkaline character, consistent with typical seafood wastewater [25].

Compared with previously reported values for seafood cooking juices [13,14,15,16,17,18,19,20,21,22,23,24,25,26], surimi wastewater [19], and fish processing wastewater [27,28], the SWW in this study had a lower protein content and relatively higher COD. According to Walha et al. [29], the composition of wastewater can vary significantly depending on factors such as product/water ratio, species processed, and treatment time. The low protein concentration observed in the present study could be attributed to the low shrimp/water ratio used. Furthermore, the conductivity and turbidity values recorded for the SWW were significantly higher than the values reported for fish processing wastewater (93.5 NTU and 2800 µS/cm, respectively [27]).

**Table 1 foods-14-02044-t001:** Main characteristics of shrimp washing wastewater and guidelines for the discharge of wastewaters from fish processing in Tunisia and the EU.

Parameter	Initial SWW	SWW After Vacuum Filtration	Tunisia	European Union [30]
pH	7.82 ± 0.1	7.79 ± 0.12	6.5–9	5.5–8.5
Conductivity (µS/cm)	6638 ± 125	6386 ± 175.25	5000	n.e
Turbidity (NTU)	593 ± 15.70	43 ± 9.25	n.e	n.e
Dry matter (%)	0.828 ± 0.050	0.78 ± 0.07	n.e	n.e
Ash (%)	0.379 ± 0.017	0.32 ± 0.02	n.e	n.e
COD (mg·L^−1^)	4624 ± 54.77	4272 ± 81.32	1000	110
TN (mg·L^−1^)	719 ± 25.94	619 ± 73.07	100	25
Crude protein (g·L^−1^)	1.686 ± 0.080	1.48 ± 0.05	n.e	n.e

n.e: Not established.

Vacuum filtration was used to remove suspended matter, resulting in a 93% SWW wastewater turbidity and a slight reduction in conductivity (4%), total solids (6%), ash content (16%), and crude protein content (12%). However, it is important to note that COD and total nitrogen concentrations remained significantly elevated, exceeding the permissible discharge limits.

### 3.2. Water Membrane Permeability and Flux Evolution During UF-DF

Before the shrimp washing wastewater was subjected to the ultrafiltration–diafiltration process, selected membranes were characterized for their water permeability with distilled water to determine intrinsic membrane resistance and restoration of initial flux after cleaning. The results are shown in Figure 1.

As expected, in the absence of fouling, water flux showed a linear relationship with transmembrane pressure over the tested range (1–5 bar), suggesting that flux was solely governed by membrane resistance. Furthermore, membranes with larger molecular weight cut-offs exhibited higher water fluxes. Specifically, at a working TMP of 2 bar (the pressure used during the SWW ultrafiltration–diafiltration), the 5, 10, and 50 kDa membranes showed fluxes of 118, 232, and 948 L·h^−1^·m^−2^, respectively. This observation is consistent with the Hagen–Poiseuille equation, which predicts a direct relationship between pore size and permeate flux, assuming consistent pore structure and distribution [31].

Figure 2 illustrates the evolution of permeate flux as a function of permeate volume collected during the ultrafiltration–diafiltration of the SWW using 5, 10, and 50 kDa membranes. During the initial concentration, total permeate flux decreases of 11.6%, 15.2%, and 28% were recorded for the 5, 10, and 50 kDa membranes, respectively. This flux reduction is consistent with previous studies on various food processing wastewaters, including seafood wastewater [32], poultry [33], and olive oil processing wastewater [34]. Such declines are primarily due to membrane fouling, which is caused by the accumulation and adsorption of foulants on and within the membrane pores, as well as concentration polarization, where the accumulation of solutes near the membrane surface hinders performance [35].

As expected, the 50 kDa membrane, with its larger MWCO, exhibited the highest initial flux (118 L·h^−1^·m^−2^) due to its larger pore size. Interestingly, despite significant differences in their initial water permeability, the 10 and 50 kDa membranes achieved similar final permeate fluxes (87 L·h^−1^·m^−2^), exceeding the values reported in some fish processing wastewater treatment studies [32,36]. However, this final permeate flux is comparable to the 83.1 L.h^−1^·m^−2^ reported for shrimp wastewater treatment using an ultrafiltration membrane [23]. The 5 kDa membrane with the lowest MWCO, showed the lowest final flux (71 L·h^−1^·m^−2^).

The largest decrease in permeate flux observed for the 50 kDa membrane was likely due to its higher permeability, which led to increased convective transport and, as a result, greater accumulation of foulants on the membrane surface. Additionally, larger pores were more susceptible to internal fouling by smaller components in the SWW, further increasing flow resistance. In contrast, the smaller pore size of the 5 kDa membrane provided a lower initial flux but more stable performance [36]. The 10 kDa membrane showed intermediate behavior, suggesting a balance between permeability and fouling resistance. These results highlight the complex interplay of factors that influence ultrafiltration performance beyond simple pore size.

The diafiltration step restored permeate flux for all membranes, highlighting the presence of reversible fouling that could be mitigated by dilution. However, as diafiltration 1 progressed, the permeate flux decreased again, albeit more slowly than in the initial concentration phase. This decrease was more pronounced for membranes with larger pore sizes. Specifically, during the first diafiltration (DF1), the total flux decreases were 6.7%, 14.0%, and 30.2% for the 5, 10, and 50 kDa membranes, respectively. A similar trend in flux decline was observed during the second diafiltration, with an overall decline of 9.7%, 14.6%, and 26.5% for the respective membranes. This secondary decline suggests that some degree of irreversible fouling also occurred during diafiltration, possibly due to pore constriction or adsorption of foulants within the membrane structure. This finding is consistent with previous studies demonstrating the limited impact of diafiltration on irreversible fouling [37]. For instance, a recent study found an overall flux reduction of 43.11% during the first diafiltration of mealworm protein concentrate, a value significantly higher than the reductions observed in our study (6.7–30.2%) [38]. The 5 kDa membrane consistently showed the lowest flux decline throughout both diafiltration steps.

Interestingly, despite its initial susceptibility to fouling, the 50 kDa membrane achieved higher permeate fluxes than the 10 kDa membrane during both diafiltration stages, in contrast to the concentration phase, where their fluxes were comparable. This suggests that the fouling in the 50 kDa membrane was predominantly reversible.

### 3.3. Membrane Fouling

#### 3.3.1. Fouling Index and Cleaning Efficiency

The fouling index (FI) and cleaning efficiency (CE) of the three ultrafiltration membranes under investigation are presented in Table 2. The FI values, which ranged from 29.51 ± 4.43% to 47.68 ± 1.07%, indicate a moderate fouling level across all membranes. The 5 kDa membrane showed the lowest FI and highest CE, indicating improved fouling resistance. This pattern is consistent with the permeate flux behavior observed during both the concentration and diafiltration phases. The 10 and 50 kDa membranes, despite their differing pore sizes, exhibited similar FI values, suggesting comparable fouling levels. This observation is supported by their similar flux reductions in the initial concentration stage. The CE data further corroborate the flux recovery trends observed during diafiltration. While diafiltration effectively reduced some fouling, the incomplete flux recovery and subsequent declines indicated the presence of irreversible fouling. The 10 kDa membrane exhibited the lowest cleaning efficiency (CE), indicating a reduced cleanability compared to other membranes. This observation is consistent with its flux performance during diafiltration, which demonstrated a significant decrease relative to the 5 kDa membrane. Although the 10 and 50 kDa membranes displayed similar fouling index (FI) values and comparable fouling levels during concentration, notable differences emerged in the diafiltration phase. The 10 kDa membrane, despite achieving a lower but more stable flux during diafiltration, exhibited the lowest CE, suggesting a higher proportion of irreversible fouling that resulted in sustained reduced permeability. In contrast, the 50 kDa membrane, which demonstrated a greater initial flux decline, exhibited higher flux during diafiltration and a larger proportion of reversible fouling, consistent with its higher CE and improved permeability recovery.

#### 3.3.2. Analysis of Resistances

The decline in permeate flux during an ultrafiltration process can be analyzed using the resistances-in-series model. According to this model, the permeate flux (*J_p_*) follows the general Darcy law:(4)$$J_{p}=\frac{∆P}{\mu R_{t}}$$
where *R_t_* represents the total resistance to filtration and is composed of several contributing resistances:R_t_ = R_m_ + R_fc_ = R_m_ + R_rev_ + R_irr_(5)
where R_m_ is the inherent hydraulic resistance of a clean membrane, and R_fc_ is the fouling resistance, which consists of external (R_rev_) and internal (R_irr_) fouling resistances. The reversible fouling resistance (R_rev_) arises from concentration polarization and the deposition of solids (cake layer) on the membrane surface. It can be removed through water rinsing after the ultrafiltration of the effluent. Conversely, the irreversible fouling resistance (R_irr_) is attributed to pore blocking and the adsorption of organic and particulate matter onto the membrane surface and within its pores. This form of fouling is more persistent and cannot be eliminated by simple water rinsing.

Table 3 summarizes the resistance values obtained for each MWCO tested membrane. These resistances were determined using steady-state permeate fluxes and pure water flux measurements.

The 5 kDa membrane exhibited the highest total resistance (R_t_), which was expected due to its tighter pore structure. Interestingly, membrane resistance (R_m_) accounted for 58.44% of the total resistance, suggesting that intrinsic structural resistance was a key factor. However, fouling was also present, as indicated by the reversible (R_rev_) and irreversible (R_irr_) resistance components. Among the tested membranes, the 10 kDa membrane showed the lowest total resistance. Membrane resistance’s contribution decreased to 45.53% with this MWCO, meaning that fouling resistance became the dominant factor. In particular, reversible fouling (R_rev_ = 36.85%) was the major contributor to the overall fouling, suggesting that while larger pores reduce intrinsic resistance, they become more susceptible to the deposition of foulants on the membrane surface. In comparison to the 5 kDa membrane, the reduced percentage of irreversible fouling suggests that deep pore blocking was less important.

The 50 kDa membrane had the highest fouling resistance (R_fc_), with reversible fouling (R_rev_) resistance making a substantial contribution (84.04%). This suggests that even though the larger pore size allows for a higher initial flux, a cake layer forms on the membrane surface as a result.

To further illustrate the distribution of reversible and irreversible fouling within the total fouling resistance (R_fc_), Figure 3 presents the ratio of R_rev_/R_fc_ and R_irr_/R_fc_ for the different membranes.

Figure 3 illustrates the impact of the molecular weight cut-off on fouling behavior. The highest percentage of irreversible fouling was seen in the 5 kDa membrane as a result of pore blockage and the adsorption of smaller organic molecules inside the membrane pores [39]. On the other hand, the 50 kDa membrane showed a higher percentage of reversible fouling, which was caused by the cake layer that formed on the membrane surface as a result of its larger pore size [40]. The predominant mechanism in the intermediate fouling profile of the 10 kDa membrane was reversible fouling.

Complex interactions between the membrane and effluent constituents control membrane fouling in ultrafiltration [41]. Proteins, lipids, suspended solids, and salts are among the various organic and inorganic components found in shrimp processing wastewater. Depending on the molecular size, charge, and affinity for the membrane material, each component contributes differently to the development of fouling.

Reversible fouling mainly arises from concentration polarization and the development of a gel or cake layer on the membrane surface [42]. Proteins, lipids, and colloidal macromolecules in the effluent build up at the membrane interface to create a permeable but resistant layer that prevents solvent transfer. Lipids and emulsions can make reversible fouling worse by promoting particle aggregation and the development of a thicker cake layer. As observed, the 50 kDa membrane, with its larger pores, was most susceptible to reversible fouling due to increased surface deposition of larger particles and macromolecules.

Irreversible fouling results from the adsorption of lipids, proteins, and smaller particles and pore blockage [43]. Low-molecular-weight proteins in particular can get into membrane pores, making it difficult to rinse them out. This phenomenon was most pronounced in the 5 kDa membrane because the smaller pore size promoted internal fouling. On the other hand, the 50 kDa membrane’s larger pore structure reduced internal adsorption and resulted in less irreversible fouling.

### 3.4. Fractions Characterization: SDS-PAGE Protein Profiles

Figure 4 shows the SDS-PAGE protein profiles of the SWW (initial effluent) and the ultrafiltration fractions with the selected membranes. The protein profile of the SWW displayed multiple bands ranging from 15 to 100 kDa, indicating the presence of a diverse range of proteins. A similar protein profile was observed for the initial SWW and the retentates. However, the bands were more intense for the retentates compared to the initial SWW. The intensity of the protein bands in the retentates followed the order of 5 kDa > 10 kDa > 50 kDa, demonstrating that membranes with lower MWCOs retained a higher concentration of proteins. Notably, no visible bands were detected in the 5 kDa permeate, confirming the effective retention of proteins by this membrane. However, for the 10 kDa and 50 kDa permeates, the presence of unexpected protein bands larger than 50 kDa suggests a potential protein aggregation allowing larger proteins to pass through. Proteins found in the SWW and UF fractions were probably derived from exoskeletal remnants, muscle tissue, or enzymatic secretions, all of which are frequently found in effluents from shrimp processing [44].

### 3.5. Solute Rejection and Retentate Composition

The performance of the selected membranes in treating SWW was evaluated based on the rejection rates of key parameters, including COD, turbidity, ash, and protein, as shown in Figure 5.

All membranes showed turbidity rejection exceeding 90%. These high percentages are explained by the fact that the majority of the turbidity-causing solids were larger than the pores of the membranes, which is consistent with the high turbidity retention typically observed in ultrafiltration membranes used to treat agro-industrial wastewaters [45,46]. However, the COD removal efficiency was impacted by the nominal molecular weight cut-off. As expected, the rejection rate generally increased as the MWCO decreased [44,45,46,47,48]. The 5 kDa membrane achieved the highest COD rejection (91.62%), while the 50 kDa membrane exhibited a lower rejection (77.33%), indicating reduced separation efficiency for larger organic molecules. This difference in COD rejection reflects the ability of the smaller-pore-size membrane to retain lower-molecular-weight organic compounds contributing to COD. It is worth noting that the final permeate COD concentrations for the 5 kDa, 10 kDa, and 50 kDa membranes were, respectively, 189.00 ± 69.64 mg/L, 357.54 ± 104.95 mg/L, and 1129.28 ± 14.27 mg/L. These findings suggest the suitability of 5 kDa and 10 kDa membranes for SWW treatment, as they generated permeates with COD levels significantly below the generally accepted safe direct discharge limit of 1000 mg/L in Tunisia. On the other hand, the COD concentration in the permeate of the 50 kDa membrane exceeded this cutoff.

The rejection rates for ash were 53.73%, 38.56%, and 27.77% for the 5, 10, and 50 kDa membranes, respectively. Given that salts have a low molecular weight, this trend seems counterintuitive. This unexpected trend can be explained by membrane fouling. Indeed, by acting as a secondary membrane, the accumulating layer of rejected material on the membrane surface can change the rejection characteristics and effective pore size. This fouling layer can adsorb ions, effectively creating a denser barrier that hinders the passage of salts, as described by Lin et al. [48]. In their study of high-saline organic wastewater, they observed retention of NaCl by a fouled UF membrane despite the fact that the membrane’s MWCO should theoretically permit free passage of the salt. Additionally, the membrane pore size can affect the fouling layer’s composition. Due to the higher initial rejection of larger molecules, which, in turn, affects the subsequent rejection of smaller components like salts, smaller-pore-size membranes may create a denser, more compact fouling layer [49]. On the other hand, a looser fouling layer may form on larger-pore-size membranes that, while still contributing to some salt rejection, is less effective than the denser layers formed on smaller-pore-size membranes.

The rejection rates of protein were 90.8%, 85.91%, and 80.66% for the 5, 10, and 50 kDa membranes, respectively. The highest rejection rate observed with the 5 kDa membrane indicates its superior ability to retain proteins due to its smaller pore size, which restricts the passage of larger protein molecules. As the MWCO increased to 10 and 50 kDa, the rejection rates decreased correspondingly, implying that more proteins, particularly those with lower molecular weights, were able to pass through the membrane [50]. These findings are consistent with the SDS-PAGE protein profile of SWW, which revealed the presence of proteins ranging from 15 to 100 kDa.

Figure 6 presents the dry-basis concentration (%) of protein and ash in the initial SWW and the final retentates after the UF-DF sequence.

For all tested membranes, the UF-DF process increased the protein content of the final retentate compared to the initial SWW, with a corresponding decrease in ash content. This indicates that the UF-DF process effectively separated protein from other components present in the SWW.

The 5 kDa membrane achieved the highest protein concentration in the retentate (63.68%), compared to 18.27% in the initial SWW, representing a concentration factor of 3.48. While this membrane also resulted in the highest ash concentration in the retentate (28.45%), this value remained lower than the initial SWW concentration (40.23%). With increasing MWCO, the protein concentration in the final retentate decreased. The 10 kDa membrane exhibited a final retentate protein concentration of 41.27%, corresponding to a concentration factor of 2.56, and the 50 kDa membrane achieved the lowest concentration in the retentate of 37.44%, corresponding to a concentration factor of 1.68. Interestingly, the ash content followed the same trend as the membrane fouling. This aligns with the high rejection coefficient.

Comparing these retentates to the initial SWW, the UF process demonstrated a significant enrichment of protein while reducing the ash content, particularly for the 5 kDa membrane, which provided the highest protein purity.

## 4. Conclusions

This study demonstrated the effectiveness of ultrafiltration–diafiltration (UF-DF) for treating shrimp washing wastewater (SWW) and recovering high-value proteins. Among the membranes tested (5, 10, and 50 kDa), the 5 kDa membrane showed the best overall performance, achieving over 90% protein retention and COD rejection, along with the highest protein concentration in the retentate (63.7% on a dry basis). Additionally, it demonstrated superior operational stability, the lowest fouling index, and the highest cleaning efficiency, making it a promising option for sustainable wastewater valorization.

These findings demonstrate that UF-DF is an environmentally friendly approach to simultaneously reduce environmental discharge and recover valuable biocomponents from seafood processing effluents. Future research should focus on scaling-up the process, evaluating the functional properties of the recovered proteins, and conducting comprehensive techno-economic and life cycle assessments to support industrial implementation.

## Figures and Tables

**Figure 1 foods-14-02044-f001:**
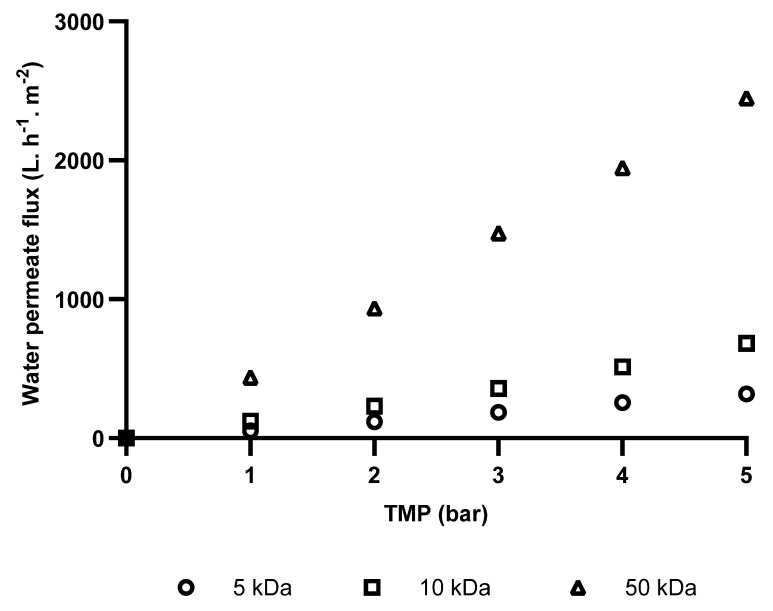
Water permeate fluxes through the tested ultrafiltration membranes for different TMP at 20 °C.

**Figure 2 foods-14-02044-f002:**
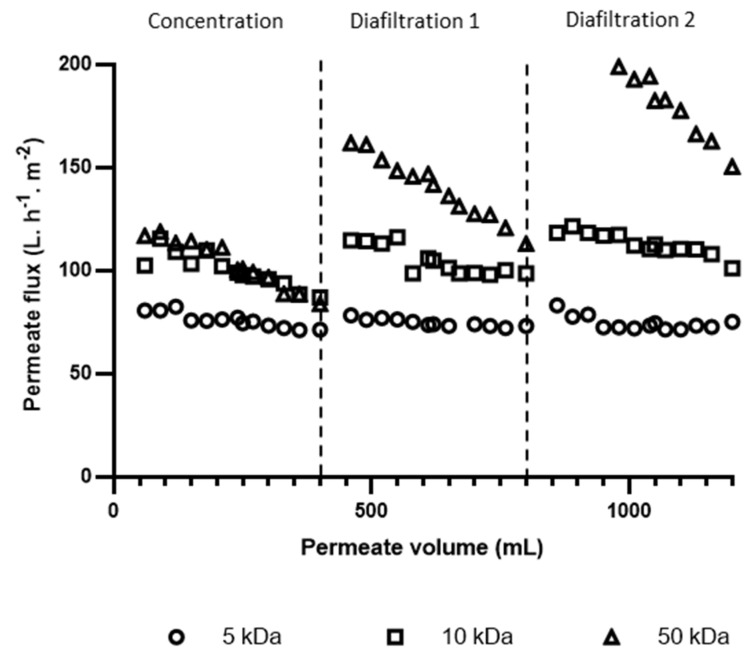
Ultrafiltration permeate flux evolution during treatment of shrimp washing wastewaters using 5, 10, and 50 kDa membranes. Data represent triplicate measurements (*n* = 3). Operating conditions: ΔP = 2 bar, T = 20 °C, v = 1.7 m·s^−1^.

**Figure 3 foods-14-02044-f003:**
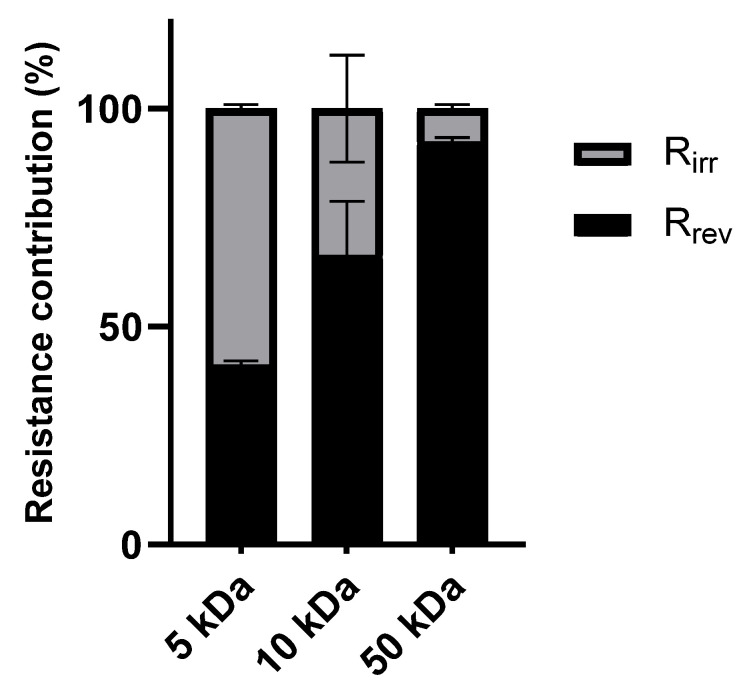
Contribution of R_rev_ and R_irr_ to the total fouling resistance (R_fc_) for the different tested membranes.

**Figure 4 foods-14-02044-f004:**
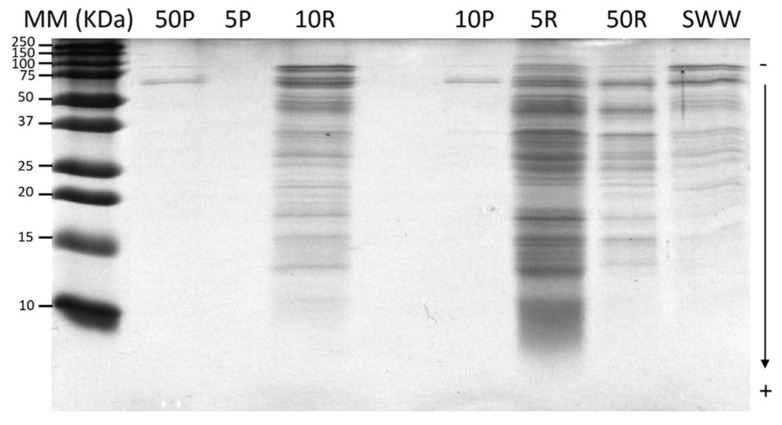
SDS-PAGE protein profiles of SWW and ultrafiltration fractions with 5, 10, and 50 kDa MWCOs. MM: molecular weight marker from 10 kDa to 250kDa, 50P: 50 kDa permeate, 5P: 5 kDa permeate, 10R: 10 kDa retentate, 10P: 10 kDa permeate, 5R: 5 kDa retentate, 50R: 50 kDa retentate, and SWW: shrimp washing wastewater.

**Figure 5 foods-14-02044-f005:**
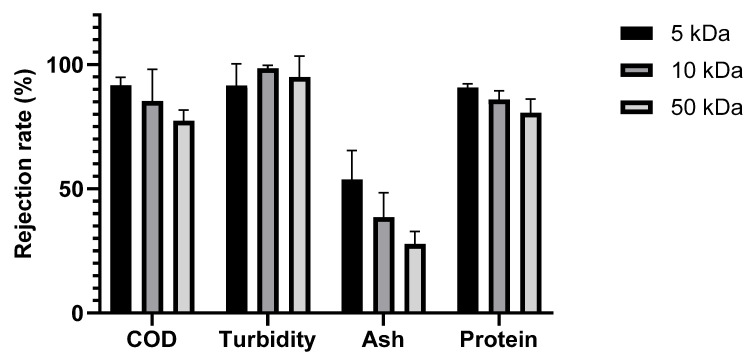
Retention of selected membranes towards chemical oxygen demand (COD), turbidity, ash and protein.

**Figure 6 foods-14-02044-f006:**
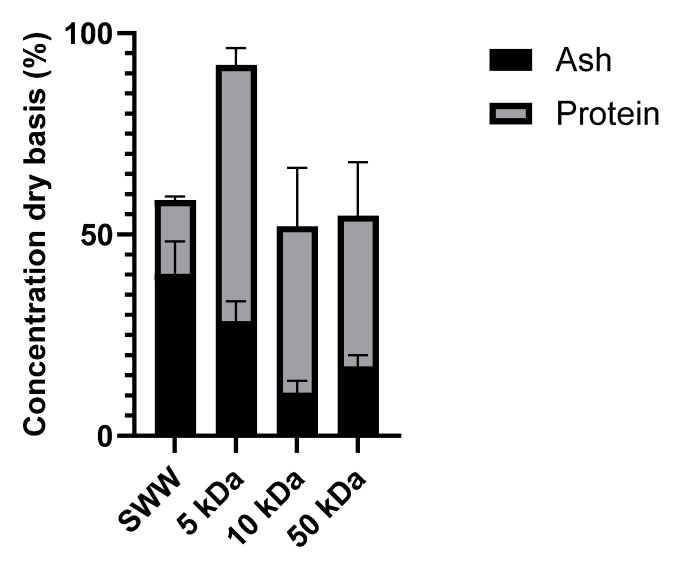
Protein and ash composition of the retentate fractions with the tested membranes at the end of the UF-DF steps.

**Table 2 foods-14-02044-t002:** Water permeabilities, fouling index, and cleaning efficiency of membranes tested in the ultrafiltration of SWW.

	Membrane MWCO		
	5 kDa	10 kDa	50 kDa
L_p0_ (×10^10^ m·s^−1^·Pa^−1^)	1.76 ± 0.01	4.04 ± 0.64	13.39 ± 0.02
L_p1_ (×10^10^ m·s^−1^·Pa^−1^)	1.24 ± 0.80	2.09 ± 0.59	7.69 ± 0.42
L_p2_ (×10^10^ m·s^−1^·Pa^−1^)	1.41 ± 0.16	2.59 ± 0.40	10.04 ± 0.02
FI (%)	29.51 ± 4.43	47.68 ± 1.07	42.53 ± 2.32
CE (%)	80.22 ± 9.34	62.26 ± 2.70	75.03 ± 2.77

**Table 3 foods-14-02044-t003:** Resistances during UF of shrimp processing effluents for the tested membranes.

MWCO (kDa)	R_t_ (×10^12^ m^−1^)	R_m_ (×10^12^ m^−1^)	R_fc_ (×10^12^ m^−1^)	R_rev_ (×10^12^ m^−1^)	R_irr_ (×10^12^ m^−1^)	R_m_/R_t_ (%)	R_fc_/R_t_ (%)	R_rev_/R_t_ (%)	R_irr_/R_t_ (%)
5	9.99	5.68	4.08	1.69	2.39	58.44	41.55	17.17	24.38
10	6.77	2.97	3.70	2.51	1.19	45.53	54.47	36.85	17.62
50	8.15	0.79	7.36	6.52	0.55	9.72	90.84	84.04	6.80

## Data Availability

The original contributions presented in the study are included in the article, further inquiries can be directed to the corresponding author.

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
