# Peer review of "Sustainable Protein Recovery and Wastewater Valorization in Shrimp Processing by Ultrafiltration"

_foods, 2025, doi:10.3390/foods14122044_

Round 1
Reviewer 1 Report
Comments and Suggestions for Authors
Dear Authors,
Your manuscript presents an important effort toward sustainable protein recovery and wastewater valorization in shrimp processing. The experimental design is scientifically sound, and the presentation of results is generally clear.
However, I noted a critical conceptual weakness in the Introduction (Page 2) where the justification for combining ultrafiltration (UF) and diafiltration (DF) for "sustainable protein recovery" is introduced. The sentence—"This study fills this significant gap in the literature by evaluating the effectiveness of ultrafiltration-diafiltration (UF-DF) process..."—does not sufficiently explain why the DF step is critical for achieving sustainability in this context. While UF is widely used, the addition of DF needs stronger justification specific to shrimp washing wastewater, especially in terms of selectivity for low-molecular-weight impurities and energy/resource efficiency.
I recommend that you:
- Clarify the environmental and technological rationale for the addition of DF to UF in this specific application.
- Better differentiate your work from prior studies such as those by Li et al. and Tonon et al.
- Consider a visual schematic summarizing the novelty and sustainability benefit of the proposed UF-DF sequence.
Minor grammatical and typographical errors (e.g., "ditribution" instead of "distribution") should also be corrected to improve the manuscript’s readability.
I look forward to a revised version that addresses these concerns.

The manuscript is generally well-written and understandable. However, a few minor typographical errors and word choices should be corrected to improve readability. Examples include misspellings such as 'ditribution' (should be 'distribution') and unnecessary duplicated words like 'Indeedd,'. These are minor and do not obstruct overall comprehension.
Author Response
Comment 1 : I noted a conceptual gap in the Introduction where the UF-DF process is introduced. While the use of UF is well supported by literature, the justification for incorporating diafiltration for improved sustainability in this specific context (shrimp washing wastewater) is weak. I suggest clarifying the rationale for DF addition and providing a stronger link to the sustainability objective.
Response1 : We thank the reviewer for this valuable suggestion. To address this point, we have revised the Introduction to provide a clearer and stronger justification for the integration of diafiltration (DF) into the UF process. We now emphasize that DF enhances the selectivity and sustainability of the process by removing low-molecular-weight impurities from the retentate, leading to improved protein purity and lower contaminant loads in the final permeate. This supports both environmental goals and resource recovery from shrimp processing effluents.
We also explicitly referenced literature supporting DF’s benefits in related applications [21–23] and highlighted the novelty of applying UF-DF to shrimp washing wastewater specifically.
The updated passage that was added to the introduction is below (paragraph 4):
« To further enhance selectivity and sustainability, diafiltration (DF) can be integrated with UF. DF involves the addition of clean water to the retentate during filtration, which helps to remove smaller undesirable solutes like salts, small organics, and low-molecular-weight impurities. This results in higher purity and concentration of target macromolecules like proteins. Importantly, by reducing the contaminant load in the final permeate and enhancing retentate quality, DF supports circular resource recovery and environmental protection [21,22]. For example, Li et al [23] demonstrated this by effectively obtaining a protease-rich concentrate from tuna spleen extract using UF-DF sequence with a 30 kDa membrane. Similarly, Tonon et al [24] further illustrated the benefits of UF pre-treatment for concentrating shrimp cooking water prior to enzymatic hydrolysis, resulting in a fraction rich in essential amino acids and antioxidants. However, despite growing evidence of its effectiveness in seafood processing, the combined UF-DF approach is still not explored for the treatment of shrimp washing wastewater specifically, a rich but under-exploited waste stream. »
Comment 2 : Clarify the definition and calculation method of 'diavolume' in Section 2.2.
Response 2 : Thank you for pointing out the need for clarity regarding the definition of "diavolume." We have revised Section 2.2 of the manuscript to explicitly define this term.
The following sentence was added to Section 2.2:
« One diavolume (DV) corresponds to the volume of deionized water added to the retentate during diafiltration that is equal to the initial volume of the retentate before dilution. »
Comment 3 : Explain SDS-PAGE sample preparation under low protein concentration in Section 2.4.
Response 3 : We have clarified the sample preparation procedure in Section 2.4 to address the handling of low protein concentrations. Additional details on buffer composition, heating, and sample loading have been added.
According to the BCA protein assay kit’s protein concentration results, all samples had low protein content (< 2 µg.µL-1). As a result, all samples were half-diluted using 125 mM Tris-HCl buffer, pH 6.8, containing 0.2% (w/v) SDS, 5% (v/v) β-mercaptoethanol, 10% (v/v) glycerol and 0.01% (w/v) bromophenol blue (100µL of sample with 100µL of buffer). Each sample was loaded into the gel with a volume of 15 µL after being heated for 10 minutes at 95°C.
Comment 4 : Add a source or explanation for the COD discharge limit mentioned in Table 1.
Response 4 : Thank you for highlighting this. We have added references to the respective regulatory frameworks to the reference list.
Comment 5 : Improve clarity in figure captions (e.g., Figure 2: triplicate data, SD bars).
Response 5 : Thank you for this useful suggestion. We have revised the caption of Figure 2 to clarify that the data represent triplicate measurements (n = 3). However, we opted not to include the standard deviation (SD) bars directly on the figure to avoid overloading the graph and compromising its readability, especially since the differences between replicates were minimal and the curves are already visually dense.
We have updated the figure caption as follows :
« Ultrafiltration permeate flux evolution during shrimp washing wastewaters treatment using 5, 10 and 50 kDa membranes. Data represent triplicate measurements (n=3). Operating conditions : ∆P = 2 bar, T = 20°C, v = 1.7 m.s-1. »
Comment 6 : Correct minor typographical errors such as 'ditribution' and 'Indeedd,'
Response 6 : We thank the reviewer for pointing this out. A comprehensive proofreading was carried out, and typographical errors such as “ditribution” (corrected to “distribution”) and “Indeedd” (corrected to “Indeed”) have been fixed. Additional minor typographic were also corrected throughout the manuscript.

Reviewer 2 Report
Comments and Suggestions for Authors
This manuscript focused on the use of ultrafiltration for sustainable protein recovery and the treatment of shrimp washing wastewater. It was a very interesting subject. The manuscript was well written. However, the manuscript leaves a lot to be improved.
1.INTRODUCTION: Please describe what are the main wastes in the shrimp industry. Preferably, they can be categorised. And which category is of major concern in this study.
2.What are the characteristics of wastewater from shrimp processing compared to other wastewater.
3.Section 2.1: What’s the meaning of “ The wastewater was collected under frozen conditions”? Were the wastewater frozen?
4.What was the purpose of determining COD?
5.Please give the full name of the BAC when it first appears.
6.How was the experimental data handled? How many parallel experiments were done? Was an analysis of variance performed?
7.The conclusion was too much and didn't summarise the results of this study very well. Please revise to 1-2 paragraphs.
Author Response
Comment 1 : Introduction: Please describe what are the main wastes in the shrimp industry. Preferably, they can be categorised. And which category is of major concern in this study.
Response 1 : We agree with the reviewer and have revised the Introduction (Section 1, paragraph 1) to include a clear categorization of shrimp processing wastes. We now specify the types of waste (solid and liquid), their composition, and highlight that our study focuses on liquid waste, specifically shrimp washing wastewater (SWW).
Revision added :
Despite its economic significance, the shrimp processing industry produces a lot of waste, which represents a substantial environmental challenge and results in the loss of valuable resources. These wastes can be roughly divided into two categories: solid wastes, such as heads, shells and exoskeletons, and liquid wastes, including washing waters and process effluents [1]. Rich in proteins, chitin, and carotenoids, solid by-products have attracted increasing attention for their value-added applications such as chitosan production, animal feed, and biofertilizers [2]. However, liquid wastes remain underutilized and environmentally problematic, due to their high volumes and pollutant loads [3]. Washing operations alone can consume 10 to 40 m3 of water per ton per ton of raw shrimp, generating effluents rich in organic matter, fats, and nutrients [4].
Comment 2 : What are the characteristics of wastewater from shrimp processing compared to other wastewater.
Response 2 : We have expanded the Introduction (end of paragraph 1) to include a comparison of shrimp processing wastewater with other types of industrial effluents, particularly in terms of COD, turbidity, and salinity.
Revision added :
Compared to other agro-industrial effluents, shrimp wastewater is uniquely challenging because it contains high concentrations of proteins, lipids, and carbohydrates from shrimp tissue and processing aids, as well as high levels of nitrogen and phosphorus from protein degradation and phosphate additives [5–7]. Additionally, it may contain chitin fragments and exhibit high salinity, especially in marine shrimp processing, further complicating treatment and disposal. These factors contribute to extremely high chemical oxygen demand (COD), which depletes oxygen in receiving waters, and high turbidity and nutrient loads that hinder photosynthesis and promote eutrophica-tion [8–10].
Comment 3 : Section 2.1: What’s the meaning of “ The wastewater was collected under frozen conditions”? Were the wastewater frozen ?
Response 3 : Thank you for pointing this out. We revised Section 2.1 for clarity. The phrase was ambiguous. We meant that wastewater samples were frozen immediately after collection to preserve their composition before ultrafiltration processing.
Revision made in Section 2.1:
“The wastewater was collected and transported under frozen conditions to the laboratory. Upon arrival, subsamples were taken for analysis, and the remaining portions were stored at -18 °C until further use. ”
Comment 4 : What was the purpose of determining COD?
Response 4 : The purpose of COD determination has been clarified in Section 2.3.3 and reiterated in the Results section. COD is a key indicator of organic pollutant load in the wastewater and is used to evaluate the efficiency of the ultrafiltration process in reducing environmental discharge.
Revision added :
« COD was measured in feed and permeate samples to evaluate the organic load in the shrimp washing wastewater before and after treatment. This parameter is crucial for assessing how well the ultrafiltration-diafiltration process reduces environmental pollution. »
Comment 5 : Please give the full name of the BAC when it first appears.
Response 5 : Thank you for catching this. We assume this refers to the BCA Protein Assay Kit, mentioned in Section 2.3.4. We have corrected the abbreviation upon first mention:
Revision made in Section 2.3.4:
“Protein concentration was determined using a Bicinchoninic Acid (BCA) Protein Assay Kit...”
Comment 6 : How was the experimental data handled? How many parallel experiments were done? Was an analysis of variance performed?
Response 6 : We have clarified this in Section 2.2 and 2.3. Statistical handling and replication details were added. Experiments were performed in triplicate and data are presented as mean ± standard deviation. While we did not perform ANOVA, we mention this in the limitations.
Revision added to Section 2.2 and 2.3:
“All ultrafiltration experiments were performed in triplicate using new membranes for each replicate. Results are reported as mean ± standard deviation. While no analysis of variance (ANOVA) was conducted, trends across replicates were consistent.”
Comment 7 : The conclusion was too much and didn't summarise the results of this study very well. Please revise to 1-2 paragraphs.
Response 7 : We appreciate the reviewer’s feedback. The Conclusion section has been revised and condensed to two focused paragraphs. It now clearly summarizes the main findings and future outlook, without repetition.
Revised conclusion :
«This study demonstrated the effectiveness of ultrafiltration-diafiltration (UF-DF) for treating shrimp washing wastewater (SWW) and recovering high-value proteins. Among the membranes tested (5, 10, and 50 kDa), the 5 kDa membrane showed the best overall performance, achieving over 90% protein retention and COD rejection, along with the highest protein concentration in the retentate (63.7% on a dry basis). Additionally, it demonstrated superior operational stability, the lowest fouling index, and the highest cleaning efficiency, making it a promising option for sustainable wastewater valorization.
These findings demonstrate that UF-DF is an environmentally friendly approach to simultaneously reduce environmental discharge and recover valuable biocomponents from seafood processing effluents. Future research should focus on scaling up the process, evaluating the functional properties of the recovered proteins, and conducting comprehensive techno-economic and life cycle assessments to support industrial implementation. »
